# Viscosity Measurement of CO_2_–Solvent Mixtures for the Study of the Morphology and Size of Crystalline Particles Obtained Using Supercritical Antisolvent Precipitation

**DOI:** 10.3390/ma16186151

**Published:** 2023-09-10

**Authors:** Anton M. Vorobei, Mikhail O. Kostenko, Olga O. Parenago

**Affiliations:** Kurnakov Institute of General and Inorganic Chemistry, Russian Academy of Sciences, 117901 Moscow, Russia; kostenko@supercritical.ru (M.O.K.); oparenago@scf-tp.ru (O.O.P.)

**Keywords:** supercritical antisolvent precipitation, binary fluids, crystals morphology, viscosity

## Abstract

The viscosity values of CO_2_–dimethylphormamide, chloroform, methanol, isopropanol, ethyl acetate, acetone, and dimethyl sulfoxide mixtures were measured at a pressure of 150 bar and a temperature of 313 K. The correlation of the mean size of levofloxacin hydrochloride and malonic acid particles precipitated using the SAS method with the viscosity of the used CO_2_–solvent mixtures is shown. The high viscosity of the mixtures leads to slower mixing of the solution and the antisolvent. Therefore, crystallization occurs at large fractions of the solvent, and as a consequence at a lower supersaturation. This causes the formation of larger particles when using more viscous solvents in SAS.

## 1. Introduction

The SAS method (supercritical antisolvent precipitation) is based on supersaturation in a solution of a micronized substance using a supercritical fluid as an antisolvent. In the vast majority of cases, supercritical carbon dioxide (SC CO_2_) acts as an antisolvent [1]. The SAS method is most widely used to obtain micro-, submicro- and nanoparticles of pharmaceutical substances [2,3,4,5]. However, recently, it has also often been used to create particles of catalyst precursors [6,7,8,9] and in various separation processes [10,11,12]. Among the approaches to obtaining microparticles based on the use of supercritical fluids, the SAS method is the most versatile, since most pharmaceutical substances are polar organic salts with very poor solubility in SC CO_2_. An important and main advantage of the SAS method is the wide possibilities to control the size and morphology of the obtained particles, which are achieved by varying the process parameters (pressure, temperature, flow rates of the solution and antisolvent, the concentration of the micronized substance in the solution, etc.). However, the large set of variable parameters makes the method more complex. An effective implementation of the SAS method is impossible without understanding the influence of its parameters on the characteristics of the resulting particles. The large number of method parameters and their complex interrelation mean that there is an urgent need for such a study [13]. 

One of the most important factors that directly determines the characteristics of the precipitated product in SAS micronization is the local supersaturation at which the process occurs. Therefore, it is necessary to determine both the thermodynamic solubility of the micronizable substance in CO_2_–solvent mixtures and take into account the specifics of mixing CO_2_ and solution during SAS [14]. In particular, to simulate the hydrodynamics of solution spraying in a supercritical fluid medium, it is necessary to determine the viscosity in the system. The solution of the described problems will make it possible to understand SAS precipitation mechanisms and increase the predictive ability of the constructed models.

The influence of the thermodynamic parameters of the SAS process on the size and morphology of the precipitated product in the preparation of amorphous particles is relatively well understood. For them, there is a qualitative understanding of the dependence of the resulting morphology on the parameters of the state of the antisolvent, which sets the competition between two processes occurring during mixing: the breakage of the solution jet and the disappearance of residual surface tension [15,16]. The effect of SAS hydrodynamic parameters on the morphology and size of amorphous particles has also been studied in sufficient detail [17,18]. Possible hydrodynamic modes of spraying are described, taking into account the parameters of nozzles, linear flow rates, and viscosity of solutions. Successful attempts to predict the mean particle size based on the results of numerical simulation of the mixing processes are demonstrated [19].

However, there are few works on micronization of crystalline particles using the SAS method [20]. Moreover, the number of micronized objects is noticeably small, as is the completeness of studying the SAS parameters for the morphology of precipitated particles, while the processes occurring during crystallization using the SAS method are noticeably more complicated than during the precipitation of amorphous particles. For a long time, it was believed that the formation of crystalline particles was characteristic only during the precipitation from the two-phase system SC CO_2_–solvent [21]. However, it was later shown that it is possible to obtain crystalline particles even with complete miscibility of the solvent and SC CO_2_ [22,23]. It was shown in many studies that the SAS method has a wide range of possibilities for varying the morphology of the resulting crystalline particles, as well as their polymorphic composition [24,25]. In particular, due to the fact that crystallization occurs very quickly during micronization using the SAS method, it opens up the possibility to obtain metastable polymorphic forms [26]. This is especially attractive for pharmaceutical applications, as different polymorphs are characterized by different dissolution kinetics [27]. Despite the fact that individual observations on the dependence of the size and morphology of crystalline particles obtained in SAS have been published and discussed in the literature, there is currently no complete understanding of the effect of various process parameters on particle formation. In particular, this is due to the complexity and, consequently, the low degree of knowledge of such basic characteristics of the system as the viscosity of mixtures of SC CO_2_–solvent.

In this work, in addition to measuring the viscosity of CO_2_–solvent mixtures, an attempt was made to find a correlation between the morphology and size of crystals precipitated using the SAS method and the measured values of viscosity. Obviously, the addition of a third component to the CO_2_–solvent system will affect the viscosity: the more pronounced, the higher its concentration. Accordingly, when possible, we tried to use sufficiently diluted solutions for micronization.

Levofloxacin hydrochloride was used as a model object, as the most common type of active pharmaceutical ingredient (organic salt). Malonic acid was also chosen as a model object, since dicarboxylic acids are widely used in the preparation of cocrystals [28,29].

Measuring the viscosity of fluids at high pressures (in particular, in the SC state) can be carried out using various types of viscometers (such as rolling ball, falling cylinder, capillary tube or vibrating wire viscometers) [30]. However, the purchase or creation of specific devices that allow working in such conditions may be impractical in the case of a relatively low intensity of their use. Several studies have previously shown the possibility of using chromatographic installations to obtain the values of the viscosity of liquids and supercritical fluids, without the need for significant equipment modifications [30,31,32]. If a suitable supercritical chromatograph is available, it is possible to measure the viscosity of fluids and their mixtures without the application of specialized equipment. The described approach makes it possible to calculate the viscosity value of chromatographic mobile phase from the pressure drop in the chromatographic column. According to Darcy’s law, the pressure drop at the ends of a porous filter medium at fluid flow is proportional to its viscosity:$$\eta =\frac{∆P\cdot A{\cdot d}^{2}}{\varphi \cdot L\cdot F}$$
where *η* is the dynamic viscosity of the fluids, ∆*P* is the pressure drop at the chromatographic column ends, *A* is the column cross-sectional area, *d* is the sorbent particle diameter, *φ* is the column resistance factor, *L* is the column length, and *F* is the volumetric flow rate of the medium in the column.

Darcy’s law is applicable under the condition of the mobile phase flow laminarity in the column. The transition of the medium flow regime from laminar to turbulent is characterized by the Reynolds number, which, for the case of fluid flow through a chromatographic column, is calculated using the formula:$${Re}_{col}=\frac{d\cdot \rho \cdot F}{\eta \cdot A\cdot \epsilon }$$
where ρ is the mean density of the medium and ϵ is the sorbent layer porosity in the column.

Experimental observations show that the transition from laminar to turbulent regime can be observed when the Reynolds number is above 1. Under analytical high-performance liquid chromatography conditions, the ${Re}_{col}$ value rarely exceeds 0.1, so the mobile phase flow through the column is usually laminar [31]. However, when switching to the case of supercritical chromatography, due to the significantly lower viscosity of the mobile phase and application of increased flow rates, the occurrence of a turbulent flow regime in the system is more probable [32].

In Ref. [32], the authors point out the occurrence of significant deviations from Darcy’s law in SFC when working with flow rates up to 5 mL/min (mobile phase—CO_2_) at an average column pressure of 150 bar and a temperature of 40 °C. In this case, the main source of nonlinearity of the pressure drop dependence from the fluid flow rate was the extra-column volumes of the system, which indicates the occurrence of turbulence during the medium flow through the capillaries, connectors, and detector cell. At the same time, when the pressure was measured directly at the ends of the column, such a dependence was almost linear, so it can be assumed that the flow is laminar, and, accordingly, the measuring of the viscosity is possible.

The direct measurement of CO_2_–cosolvent mixture viscosity using SFC is well described by Fields et al. [30]. The authors point out the presence of random and systematic errors when measuring the pressure drop and volumetric flow in the system, since a commercial chromatograph was used in the experiments without additional modifications (flow meter, direct pressure sensors at the ends of the column). This led to a discrepancy in the viscosity values obtained for the same fluid composition, but measured at different specified flow rates of the mobile phase. In general, the authors note the convenience of the chromatographic approach for the estimation of the viscosity of two or more component fluids, provided that there is no phase separation.

In addition to turbulence, the non-linearity of the pressure drop dependence from the flow rate can be influenced by the compressibility of the filter medium (adsorbent of the chromatographic column) at high flow rates of the mobile phase and some other factors [32], but their occurrence is very unlikely in classical SFC modes.

Thus, SFC can be used to measure the fluid mixture viscosities under high pressure, but it is necessary to carefully monitor compliance with Darcy’s law (the proportionality of the pressure drop in the column to the flow rate of the medium).

## 2. Materials and Methods

### 2.1. Materials

Supercritical antisolvent precipitation of levofloxacin hydrochloride (Sansh Biotech Pvt Ltd., New Delhi, India) and malonic acid (99%, Acros Organics, Shanghai, China) was carried out. The solvents used were methanol for HPLC 99.9% (Lab-Scan, Warsaw, Poland), isopropanol for chromatography (Component-Reaktiv, Moscow, Russia), extra-pure acetone (Khimmed, Moscow, Russia), 99.5% dimethyl sulfoxide (DMSO) (Lab-Scan, Warsaw, Poland), chemical pure dimethylformamide (DMF) (Khimmed, Moscow, Russia), ethyl acetate for spectroscopy (Component-Reaktiv, Moscow, Russia), and chemical pure chloroform (Khimmed, Moscow, Russia). In this work, we used food grade CO_2_ 99.5% (GOST 8050-85, Linde Gaz Rus JSC, Balashikha, Russia) as the main component of the studied mixtures.

### 2.2. Methods

#### 2.2.1. Installations for Viscosity and Density Measurements

Viscosity measurements were carried out on the installation (Figure 1), based on the supercritical fluid chromatograph Acquity UPC2 (“Waters”, Milford, MA, USA), in the following configuration:CO_2_ and co-solvent pump Acquity ccBSM;Back pressure regulator Acquity Convergence Manager.

**Figure 1 materials-16-06151-f001:**
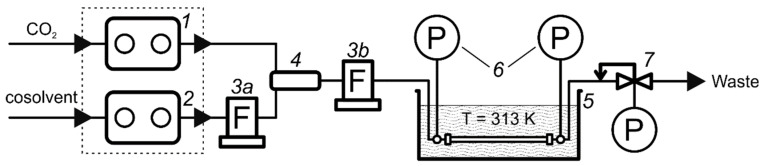
Schematic diagram of the viscosity measurement assembly: 1—CO_2_ pump, 2—solvent pump, 3—Coriolis flow meter, 4—static flow mixer, 5—chromatographic column in the liquid thermostat, 6—pressure sensors, 7—back pressure regulator.

Empower 3 (version number 3471, “Waters”, Milford, MA, USA) software was used for instrument control and data processing.

The air column thermostat, which was applied in standard chromatograph configuration, was replaced with a liquid one for greater uniformity and accuracy of maintaining the column temperature during experiments. In this work, an immersion thermostat M02 (“Thermex”, Kemerovo, Russia) was used, with a self-made 10 L water bath. To reduce the extra-column hydraulic resistance of the installation, the autosampler and diode-array detector were excluded from the chromatograph configuration, and the capillary length was also minimized.

A Luna C18-2 chromatographic column (Phenomenex, Torrance, CA, USA) with dimensions of 4.6 × 150 mm was used in this work; the diameter of the sorbent particles in the column was 3 µm. The pressure drop on the column was measured by a pair of APZ-3420 electric pressure transducers (PIEZUS, Moscow, Russia) connected to the line using Valco ZT1C tees (VICI AG International, Schenkon, Switzerland). The tees were located as close as possible to the column ends to minimize the influence of the hydraulic resistance of the connecting capillaries on the measured pressure drop.

A mini CORI-FLOW M12 Coriolis flow meter (Bronkhorst High-Tech B. V., Ruurlo, The Netherlands) was used to measure mass flows in the chromatograph system. Determining the fluid viscosity value from the Darcy equation requires the use of the volumetric flow parameter. To convert the measured values of mass flows into volumetric ones, the experimentally found values of the CO_2_–cosolvent mixture densities were used. The mixture density was measured using a pycnometric method on a self-assembled installation (Figure 2).

#### 2.2.2. Measurement of the Mixture Density

A 20 mL autoclave connected to a needle valve by a steel capillary was installed on a WA50002Y laboratory scale (W&J Instrument Co., Ltd., Changzhou, China) and weighed, then the required mass of the selected solvent was placed into the autoclave. The autoclave lid was closed, and assembly was connected through a valve to a Supercritical-24 CO_2_ pump (SSI Teledyne, State College, PA, USA). The autoclave was immersed to a thermostatic bath and kept for 5 min to reach the required temperature, after which the valve was opened and carbon dioxide was supplied to the system at a rate of 3 mL/min until the required pressure was reached. During the CO_2_ pumping, the mixture in the autoclave were regularly mixed manually by shaking with a steel stirring ball. The system was kept under such conditions for 20 min for complete CO_2_ and solvent mixing and establishment of thermodynamic equilibrium in the system. The valve was closed and disconnected from the pump, after which the autoclave filled with mixture was removed from the thermostat, wiped dry, and weighed on a balance. Since at 150 bar and 40 °C all the studied CO_2_–organic solvent systems are single-phase in any concentration range [33,34,35], based on the masses of the empty (m_0) and filled (m) autoclave, as well as its volume (V), the density of the medium was calculated with the following formula: *ρ* = (m − m_0)/V. Before the next experiment, the valve was opened and the medium was discharged from the autoclave into a special collector connected to the exhaust ventilation, then the autoclave was thoroughly cleaned of any solvent residues.

#### 2.2.3. Measurement of the Mixtures Viscosity

Due to the significant dependence of the sub- and supercritical fluids density on pressure and temperature, the actual values of the volumetric flows of CO_2_–solvent mixtures in the column may differ from those set in the program. This leads to the necessity of rigorous mass flow control of both CO_2_ and liquid solvents in the chromatographic system, followed by the calculation of the volume flow for specific conditions.

We had only one flow meter at our disposal; for this reason, the experiments were carried out twice in two configurations of the installation. In the first configuration (3a of Figure 1), a flow meter was installed in line between the liquid solvent pump and the flow mixer to measure the solvent flow. In the second configuration (3b of Figure 1), a flow meter was installed after the flow mixer to measure the total flow. This made it possible to obtain data on the real molar composition of the CO_2_–solvent mixtures supplied to the column, as well as to control the reproducibility of the results.

Before performing the main experiments on the viscosity measurement, it was necessary to make sure that the flows were laminar under the studied conditions. For this purpose, with different compositions of the CO_2_–solvent mixtures, graphs of the pressure drop (Δ*P*) dependence on the mean volumetric flow in the column were plotted. The deviation of such dependences from linearity indicates the occurrence of flow turbulence in the system. In all experiments, the average column pressure was kept constant (equal to 150 bar) by manually adjusting the back pressure at the outlet of the system.

Experimental procedure:The liquid thermostat was turned on and the temperature value was set to 313 K. The chromatograph was prepared: the pump block and the back pressure regulator were turned on and the liquid pump was washed with the selected solvent.The flow meter was connected to position 3a (Figure 1) and the system was washed with a stream of pure CO_2_.Using the control program, the volume content of the solvent in the mixture was varied in the range of 0–100% with a step of 2.5–10%. At each step, the system was maintained for about 15 min to enter the stationary mode (constant readings of pressure sensors at the ends of the column). Then, the back pressure at the outlet of the system was manually adjusted to obtain an average pressure in the column of 150 bar; the values of liquid solvent mass flow and column pressure drop were recorded.The system was stopped, the flow meter was switched to position 3b (Figure 1), and the measurements were repeated similarly to point 3, but the values of the mixture total flow in the system were recorded.

The processing of experimental data consisted in the determination of unknown parameters of the Darcy equation. The pressure drop was obtained directly from the experimental data, the cross-sectional area and column length, and the diameter of the sorbent particles were taken from the passport data. The average volumetric flow of the mixture was calculated from the total mass flow (${\dot{m}}_{tot}$) and the average density ($\rho_{av}$) of the medium in the column: $F_{av}={\dot{m}}_{tot}/\rho_{av}$. The column resistance factor was determined via the Darcy equation using known viscosity data for pure CO_2_, which were taken from the NIST Chemistry Webbook database [36]. With all the necessary data, the mixture viscosity was obtained from the Darcy equation.

#### 2.2.4. Supercritical Antisolvent Precipitation

SAS experiments were carried out using a RESS/SAS setup manufactured by Waters Corporation. The laboratory apparatus used for the SAS process is represented schematically in Figure 3.

The procedure of supercritical antisolvent precipitation using this apparatus is described in detail elsewhere [37,38]. Micronization was carried out as follows. Solutions of micronized substances were pre-prepared in different organic solvents. Levofloxacin was micronized using methanol, acetone, DMSO, DMF, ethyl acetate, and chloroform; malonic acid was micronized using isopropanol, acetone, and ethyl acetate. An ultrasonic bath was used to increase the dissolution rate. The operating parameters of pressure, temperature, and CO_2_ flow rate were set until they reached the regime values. Then, liquid pump 7 was filled with a pure solvent, and 5 mL of the solvent was sprayed at a working speed into settling vessel 8 in order to balance the composition of the fluid in the vessel. After that, the inlet line of the pump was moved into a container with solution 6 and the solution was sprayed at the same volumetric flow rate. After spraying the solution, an additional 10 mL of pure solvent was added in order to flush the solution line. At the end of the spraying, the flow of CO_2_ was additionally kept in the precipitation vessel 8 for 30 min to wash out the residual organic solvent from the product. Then, the flow of CO_2_ was stopped and automatic pressure regulator 9 was used to gradually release the pressure to an atmospheric level. The micronized powder was removed from precipitation vessel 8 using a basket built into it. The experiments were carried out at a CO_2_ flow rate of 50 g/min (according to the data of a mass flow meter), a pressure of 150 bar, a temperature of 40 °C, and a solution flow rate of 1 mL/min. The spray nozzle diameter was 100 µm.

#### 2.2.5. Scanning Electron Microscopy

Electron microscopy was performed using a LEO 1450 scanning electron microscope (SEM) (Carl Zeiss, Göttingen, Germany). The accelerating voltage was 1 kV. The sample was put on a carbon conductive bilateral adhesive tape pasted on a copper–zinc sample holder. Samples were covered by a layer of gold with a 2.5 nm thickness using the magnetron sputtering method. This procedure was performed using Quorum Q150R ES in vacuum. To achieve the limiting resolution in the vacuum chamber in which the samples were placed, the pressure was less than 5 × 10^−6^ mbar. Size analysis of micronized particles was performed using IP3 software (version number 1278, Demotech, Essen, Germany). The visible size was measured on several SEM images for each distinguishable particle. Throughout this paper, by “particle size”, we mean the particle length in its longest dimension. The arithmetic mean particle size was calculated from the data obtained.

## 3. Results and Discussion

### 3.1. Density and Viscosity of CO_2_–Solvent Mixtures

According to the previously described method, a set of isobaric–isothermal dependences of the CO_2_–solvent mixtures density on their composition was obtained in the present work; the results are presented in Figure 4. The absolute values of the density point errors did not exceed ±0.005 g/mL. The measured density values were interpolated using a smoothed cubic spline for further use in calculating the average volumetric flow rate in the column.

Experiments have shown that, under the conditions considered in the work, the fluxes of CO_2_–solvent mixtures in the column remain laminar at average flow rates of more than 1 mL/min. A demonstration of the validity of the Darcy’s law application is shown in Figure 5 on several examples of fluid compositions.

In experiments measuring the viscosity of mixtures, the average flow rates of mixtures along the column were usually 1 ± 0.1 mL/min. The design features of the back pressure regulator in the installation did not allow setting the output pressure in the system below 103.5 bar; this limited the maximum pressure drop in the system to approximately 90 bar when working with an average pressure in the column of 150 bar. For this reason, in some cases, when working with more viscous mixtures, lower flow rates (down to 0.5 mL/min) were used if the pressure drop exceeded 90 bar. Thus, all measurements were carried out in the laminar flow regime, which made it possible to use the Darcy equation. The results of the viscosity calculation for the mixtures described in this paper are shown in Figure 6.

The experiments showed the good reproducibility of the measured pressure drops. Two series of experiments performed with an interval of one week gave results differing by no more than 1%.

The viscosity curves for DMSO and isopropanol were not calculated in the entire range of solvent concentrations due to the pressure drop increase of more than 90 bar, even at mobile phase flow rates reduced to 0.5 mL/min. A further decrease in the flow rate led to a dramatic increase in flow pulsations during pump operation, and it was impossible to make reliable measurements. When working with high concentrations of DMFA, high pulsations of the mixture flow rate were also observed; for this reason, viscosity measurements were not carried out in this case.

### 3.2. Size and Morphology of Micronized Particles

Figure 7 shows SEM photos of levofloxacin hydrochloride, micronized using various solvents. The concentration of levofloxacin in the solution was 6.25 g/L. The particle size of levofloxacin micronized using the SAS method as well as the aspect ratio are demonstrated in Table 1.

As can be seen from the data in Table 1 and Figure 6, there is a correlation between the mean particle size of the precipitated particles and the viscosity of the CO_2_–solvent mixtures from which the precipitation is carried out. When using solvents with a higher viscosity (DMF), particles are formed an order of magnitude larger than with less viscous solvents (chloroform, ethyl acetate, acetone). An exception is methanol. When we use this solvent, elongated particles with a length of about 18 µm are formed. A possible explanation of this fact is given below.

Such a correlation may be due to different rates of mixing of CO_2_ and solution in the case of solvents with different viscosities, which causes the local degree of supersaturation. The saturation at which the crystallization process occurs is the main factor that determines the particle size in SAS micronization. The size of the precipitated particles is determined by the ratio of the rates of nucleation and crystal growth. The crystal growth rate is proportional to supersaturation, while the nucleation rate has a power law dependence on supersaturation. Consequently, at a low degree of supersaturation, the nucleation rate is low and active crystal growth occurs in a small number of crystallization centers. This leads to the formation of large crystals. At a high degree of supersaturation, the nuclei are formed much faster, which contributes to the formation of a larger number of smaller crystals.

Thus, with a higher viscosity of the solvent, mixing occurs more slowly, which causes the start of crystallization at large percentages of the solvent. The solubility of levofloxacin is higher at higher percentages of solvent in the CO_2_–solvent-micronized substance mixture. This leads to crystallization at lower supersaturation, and, accordingly, causes smaller-sized precipitated particles.

The fact that when using methanol, which is comparable in viscosity to, for example, chloroform, the formation of much more elongated particles takes place can be associated with the significantly higher solubility of levofloxacin in CO_2_–methanol mixture compared to other solvents used. Thus, in the case of methanol, the micronization yield (percentage of the mass of the collected powder to the mass of the micronized substance dissolved in the initial solution) was below 40%. In the case of other solvents, it was always above 80%. Thus, despite the rapid mixing of CO_2_ and the methanol solution, the degree of supersaturation during crystallization was low. It is also noteworthy that the aspect ratio of micronized particles in the case of methanol was also significantly higher compared to other solvents (Table 1).

When DMSO is used as a solvent and the levofloxacin concentration is 6.25 g/L in the initial solution, there is no precipitation of the micronized substance due to its complete dissolution in the CO_2_–DMSO mixture. Therefore, a series of experiments were carried out at a higher concentration of 25 g/L.

Table 2 shows that DMSO, which is the most viscous of all the solvents used, is also characterized by the formation of large particles with a size of about 7 μm. This fact does not contradict the hypothesis described above. In addition, it can be noted that for all solvents, a decrease in particle size is observed with increasing concentration. This dependence can be explained by an increase in supersaturation during crystallization with an increase in the concentration of the micronized substance in the initial solution.

In the case of malonic acid, the choice of solvents for micronization was more limited. Malonic acid is more soluble in the studied CO_2_–solvent mixtures, and it is not precipitated when using DMSO, DMFA, acetone and methanol. The solubility of malonic acid in chloroform is too low to use such a solution. Figure 8 shows malonic acid particles obtained using ethyl acetate (Figure 8a), acetone (Figure 8b) and isopropanol (Figure 8c) as solvents. The concentration of malonic acid in the initial solution was 75 g/L.

Figure 8 clearly demonstrates that the particle size obtained by using isopropanol as a solvent is an order of magnitude larger than in the case of acetone and ethyl acetate. The smallest particles are formed when acetone is used as a solvent. The results obtained are fully correlated with the hypothesis described above. The viscosity of the CO_2_–isopropanol mixture at a molar fraction of 75% isopropanol is approximately 2–3 times higher compared to the viscosity of the CO_2_–ethyl acetate and CO_2_–acetone mixtures, respectively. Thus, the dependence found is not specific to organic salts but is also observed in the case of other micronized substances (dicarboxylic acids).

## 4. Conclusions

Viscosities of CO_2_–dimethylformamide, chloroform, methanol, isopropanol, ethyl acetate, acetone, and dimethyl sulfoxide mixtures were measured at a pressure of 150 bar and a temperature of 40 °C. A correlation between the mean particle size of levofloxacin hydrochloride and malonic acid precipitated using the SAS method and the viscosity of the used CO_2_–solvent mixtures was demonstrated. The high viscosity of the mixtures likely leads to slower mixing of the solution and antisolvent. Thus, crystallization occurs at high solvent percentages and, as a consequence, at lower supersaturation. At low supersaturation, the growth of crystals on a small number of crystallization centers prevails; this causes the formation of larger particles during micronization using the SAS method. However, if the solubility of a substance in a CO_2_–solvent mixture is high and even if the solvent has a low viscosity, crystallization occurs at a low supersaturation degree despite the rapid mixing of antisolvent and solution. As a consequence, elongated crystals are formed.

## Figures and Tables

**Figure 2 materials-16-06151-f002:**
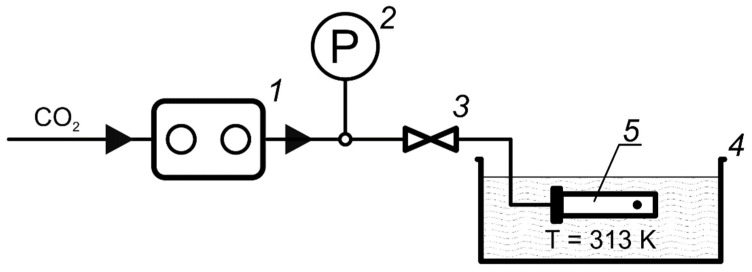
Schematic diagram of the pycnometric installation: 1—CO_2_ pump, 2—pressure gauge, 3—needle valve, 4—liquid thermostat, 5—autoclave with a stirring ball inside.

**Figure 3 materials-16-06151-f003:**
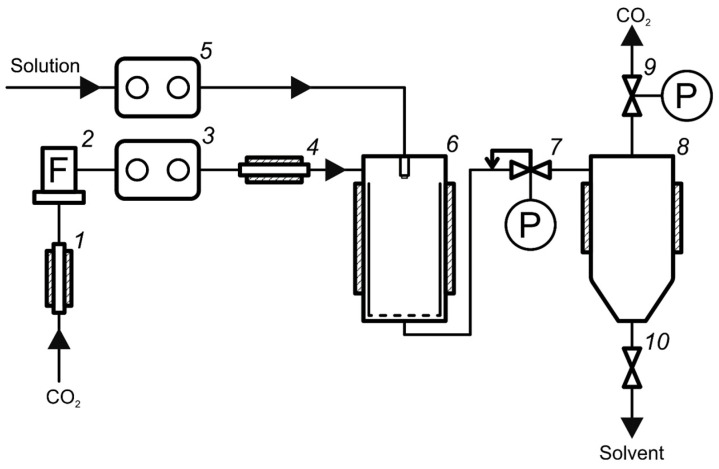
Schematic representation of SAS experimental apparatus. 1—CO_2_ cylinder; 2—flowmeter; 3—CO_2_ pump; 4—heat exchanger; 5—solution pump; 6—precipitator; 7—automatic back pressure regulator; 8—separator; 9—manual back pressure regulator; 10—valve.

**Figure 4 materials-16-06151-f004:**
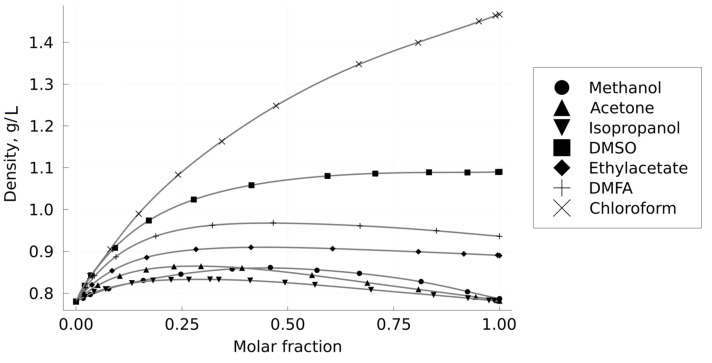
Dependences of the CO_2_–solvent mixture density on the solvent molar fraction at 313 K and 150 bar.

**Figure 5 materials-16-06151-f005:**
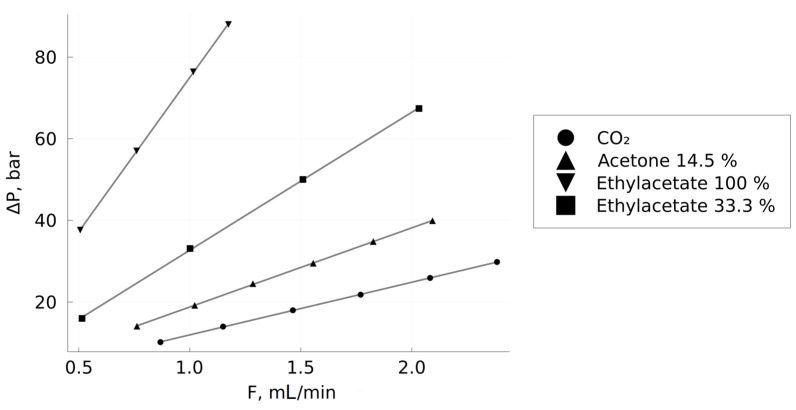
Dependences of the column pressure drop on the volumetric flow rate of CO_2_–solvent mixtures.

**Figure 6 materials-16-06151-f006:**
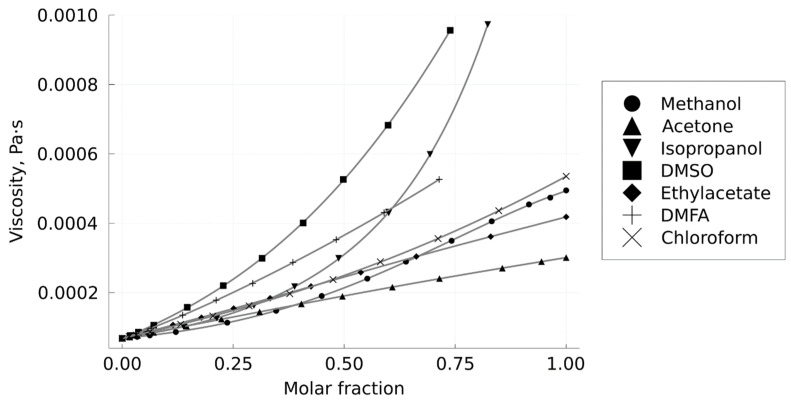
Viscosity of CO_2_–solvent mixtures at 313 K and 150 bar.

**Figure 7 materials-16-06151-f007:**
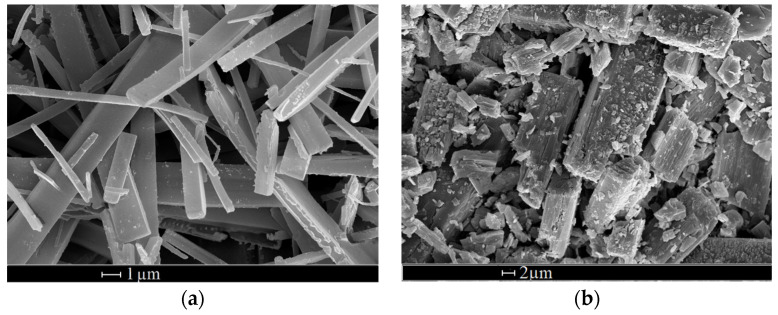
SEM photos of levofloxacin particles obtained using the SAS method using different solvents: methanol (**a**), DMF (**b**), chloroform (**c**), ethyl acetate (**d**), and acetone (**e**).

**Figure 8 materials-16-06151-f008:**
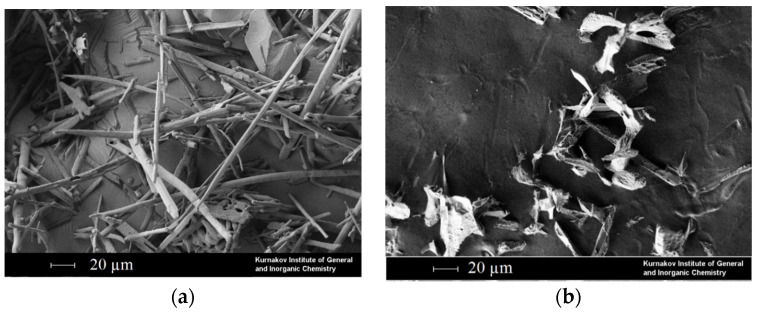
SEM photos of SAS micronized malonic acid using different solvents: ethyl acetate (**a**), acetone (**b**) and isopropanol (**c**,**d**).

**Table 1 materials-16-06151-t001:** Size and aspect ratio of levofloxacin particles obtained using the SAS method. The concentration of levofloxacin in the solution was 6.25 g/L. The solvents are arranged in decreasing viscosity.

Solvent	Mean Particle Size, µm	Aspect Ratio
DMF	31 ± 2	3.1
chloroform	2.8 ± 0.2	1.6
methanol	18 ± 1	6.1
ethyl acetate	1.6 ± 0.1	1.2
acetone	3.3 ± 0.2	1.5

**Table 2 materials-16-06151-t002:** Size and aspect ratio of levofloxacin particles obtained using the SAS method. The concentration of levofloxacin in the solution was 25 g/L. The solvents are arranged in decreasing viscosity.

Solvent	Mean Particle Size, µm	Aspect Ratio
DMSO	6.7 ± 0.5	2.1
DMF	7 ± 1	2.8
chloroform	1.4 ± 0.4	1
methanol	3.7	4.1

## Data Availability

These data can be found only in this article.

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
