# Peer review of "Viscosity Measurement of CO_2_–Solvent Mixtures for the Study of the Morphology and Size of Crystalline Particles Obtained Using Supercritical Antisolvent Precipitation"

_materials, 2023, doi:10.3390/ma16186151_

Round 1

Reviewer 1 Report

General note:

There exist 3 different phases of CO2 in the supercritical quadrant defined by p > p_crit and T > T_crit, namely, (1) pure liquid phase at 100% molar fraction with a higher density, separated by a higher-oder percolation transition from (2) supercritical colloid mesoophase with density varying linearly on molar fraction of liquid, separated in turn by another percolation transition from (3) pure gaseous phase at 100% molar fraction, with a lower density. Therefore, phase transitions may be happening e.g. along the chromatographic column in the currently reported viscosity measurements, additionally affecting the results. In particular, 313 K and 150 bar used in the measurements correspond to the liquid phase (1), with lower pressures along the column probably corresponding to mesophase (2) or even gas (3). Possible effects of these phase transitions upon the reported values of viscosity should be discussed. Reference: “Supercritical Fluid Gaseous and Liquid States: A Review of Experimental Results” I. Khmelinskii, L.V. Woodcock, Entropy 22, 437 (2020), https://doi.org/10.3390/e22040437  

Some technical questions:

326, 327, 328  90 bar when working with an average pressure in the column of 150 bar. For this reason, in some cases, when working with more viscous mixtures, lower flow rates (down to 0.5 ml/min) were used if the pressure drop exceeded 90 bar.

-- with such large pressure drop part of the column could be in under-critical conditions. does this affect the viscosity results?

-- pressure drop implies varying pressure and density along the column, and probably also viscosity varying along the column? possible effects of these pressure/density/viscosity distributions on the measured average viscosity values should be discussed.

-- maybe a shorter chromatographic column could be used, enabling measurements of higher visosities, and reducing the effects of pressure/density/viscosity distributions along the column?  

370: it would be useful to have (CO2+solvent) viscosity listed in Table 1 and Table 2

Please use consistent name for Khimmed:

151 (Khimmed, Russia), ethyl acetate for spectroscopy (Component-Reaktiv, Russia), chemi-

152 cal pure chloroform (Himmed, Russia). In the work we used food grade CO2 99.5% (GOST

Please use proper exponential form for the pressure:

297 5 × 10-6 mbar.

Some minor suggestions follow, to improve English:

137 provided that there is no phase separation occurred. -->provided that there is no phase separation occurring. (2 verbs)

142 the fluid mixtures viscosities --> the fluid mixture viscosities

144 of the pressure drop in the column from the flow rate of the medium --> of the pressure drop in the column to the flow rate of the medium

195 Measurement of the mixtures density --> Measurement of the mixture density

198 placed to the autoclave. --> placed into the autoclave.

201 to reach the required temperature by its body, --> to reach the required temperature,

208 dried from water, --> wiped dry,

214 cleaned from solvent residues. --> cleaned of any solvent residues.

237 Experiment procedure: --> Experimental procedure:

259 data, the mixtures --> data, the mixture

392 at high percentage of solvent in a mixture of CO2-solvent-micronized substance is higher. --> is higher at higher percentage of solvent in the CO2-solvent-micronized substance mixture.

444 Thus, the dependence found is not specific specifically for organic salts, --> Thus, the dependence found is not specific to organic salts,

Author Response

Thank you so much for your review and comments!

General note:

There exist 3 different phases of CO2 in the supercritical quadrant defined by p > p_crit and T > T_crit, namely, (1) pure liquid phase at 100% molar fraction with a higher density, separated by a higher-oder percolation transition from (2) supercritical colloid mesoophase with density varying linearly on molar fraction of liquid, separated in turn by another percolation transition from (3) pure gaseous phase at 100% molar fraction, with a lower density. Therefore, phase transitions may be happening e.g. along the chromatographic column in the currently reported viscosity measurements, additionally affecting the results. In particular, 313 K and 150 bar used in the measurements correspond to the liquid phase (1), with lower pressures along the column probably corresponding to mesophase (2) or even gas (3). Possible effects of these phase transitions upon the reported values of viscosity should be discussed. Reference: “Supercritical Fluid Gaseous and Liquid States: A Review of Experimental Results” I. Khmelinskii, L.V. Woodcock, Entropy 22, 437 (2020), https://doi.org/10.3390/e22040437

Potentially, the transitions described by you can significantly affect the measured values of viscosity. However, it is known that even at the lowest pressure in the considered systems (105 bar), all the chosen CO2/solvent mixtures are in the single-phase region of the phase diagram []. For most of the mixtures, we also observed the appearance in the viewing cell, and did not detect noticeable fluctuations in density and refraction, which are associated with the effects you describe. We also use similar mixtures as eluents in supercritical fluid chromatography, while any formation of inhomogeneities in the medium (for example, the mentioned colloidal mesophase) seriously affects the signal-to-noise ratio of the spectrophotometric detector. We can say with full confidence that we did not observe excessive noise on chromatograms under the considered conditions.

Technical questions:

  1. 326, 327, 328 90 bar when working with an average pressure in the column of 150 bar. For this reason, in some cases, when working with more viscous mixtures, lower flow rates (down to 0.5 ml/min) were used if the pressure drop exceeded 90 bar.

With such large pressure drop part of the column could be in under-critical conditions. does this affect the viscosity results?

With the considered pressure drop (90 bar), the pressure at the column entrance was not higher than 195 bar, at the outlet – not lower than 105 bar. For pure CO2, this entire range lies significantly above the critical point. For CO2/solvent mixtures, the phase state also depends on the molar fraction of the solvent (it can be liquid or supercritical), however, according to [], all the selected compositions locate in the single-phase region. Even when moving beyond the formal boundaries of the supercritical state existence, there will be no noticeable effect on the results provided there are no phase transitions.

Since there are no first or second order phase transitions during the transition from the liquid or vapor regions in the phase diagram to the formal region of the supercritical fluid, the curves of viscosity dependence on temperature or pressure are smooth. Examples of such curves for CO2 are presented below (NIST data).

Рressure drop implies varying pressure and density along the column, and probably also viscosity varying along the column? possible effects of these pressure/density/viscosity distributions on the measured average viscosity values should be discussed.

We agree with you, the density and viscosity vary along the length of the column, for this reason in our work we obtained some averaged approximate values of the mixtures viscosity. The curves of viscosity versus pressure in isothermal conditions are usually nonlinear for compressible media, which can lead to errors in calculating the average viscosity, especially with large pressure drops. In this paper, we neglected such errors, since the goal was only a qualitative assessment of the effect of the viscosity parameter on the size and morphology of the particles obtained by the SAS method. However, there is no doubt that the investigation of the density and viscosity profiles of CO2/solvent mixtures in the column itself is interesting from the viewpoint of describing the processes occurring in supercritical fluid chromatography, since it is known that these parameters affect the retention of substances on the sorbent.

Since the pressure drop varies linearly along the length of the chromatographic column (under laminar flow), the fluid density at each point of the column can be estimated by having a density dependence on the parameters P, T, x, for example, from the equation of state for a CO2/solvent mixture. Similarly, it is possible to estimate its distribution along the length of the column having the dependence of viscosity on the system parameters. Thus, for a detailed analysis of the viscosity determination error, an additional large-scale study is required.

Maybe a shorter chromatographic column could be used, enabling measurements of higher visosities, and reducing the effects of pressure/density/viscosity distributions along the column?

The use of shorter columns and columns with a larger sorbent particles really reduces the pressure drop and, accordingly, the density and viscosity variation along the column. However, with a decrease in the pressure drop, the relative error of its determination will significantly increase, since the error of the pressure transducers is ± 0.5 bar. In addition, pressure pulsations at the column inlet during pumps operation also amount to ± 0.2 bar. Thus, in our work we tried to maintain a balance between the accuracy of pressure registration and the magnitude of the error due to the density and viscosity gradient across the column.

  1. It would be useful to have (CO2+solvent) viscosity listed in Table 1 and Table 2

We intended to do this, however, since in SAS the CO2/solvent ratio varies along the length of the precipitation vessel, it is not clear which viscosity value to indicate in the table, for this reason we chose the current table format.

In accordance with the grammatical edits indicated by the reviewer, corrections were made to the paper.

Reviewer 2 Report

In this manuscript, the authors tried to measure the viscosity of CO2–solvent mixtures by a supercritical chromatograph, and to find a correlation between the morphology and size of crystals precipitated by the SAS method and the measured values of viscosity. However, the conditions in supercritical chromatograph and SAS method were different, for example, the mode for CO2 input was different. What's more, there doesn't seemed to have any clear correlation between solvent viscosity and the size/morphology of the crystals. Why not try to make comparison under same solubility?

Please double-check the spelling and grammar to avoid errors.

Line 444, the dependence found is not specific specifically...

Author Response

Thank you so much for your review and comments!

In this manuscript, the authors tried to measure the viscosity of CO2–solvent mixtures by a supercritical chromatograph, and to find a correlation between the morphology and size of crystals precipitated by the SAS method and the measured values of viscosity. However, the conditions in supercritical chromatograph and SAS method were different, for example, the mode for COinput was different.

SAS was performed at a pressure of 150 bar and a temperature of 40 °C, as described in Chapter 2.2.4. The temperature and average pressure in the chromatographic column were the same. The ratio of CO2 to solvent when measuring viscosity varied in the range from 0 to 1 for most solvents. When solution is sprayed in SAS, the local concentration in the CO2-solvent mixture drops from 100% solvent to the value determined by the ratio of the CO2 and solvent flow rates. The concentrations studied by chromatography in CO2-solvent mixtures covered this range for most of the solvents.

What's more, there doesn't seemed to have any clear correlation between solvent viscosity and the size/morphology of the crystals. Why not try to make comparison under same solubility?

This work was the first step, which was intended to show the correlation of the morphology of the precipitated particles and the viscosity of the mixtures used. In the future, we plan exactly such studies: comparison under same solubility.